# A Velocity Stretch Reflex Threshold Based on Muscle–Tendon Unit Peak Acceleration to Detect Possible Occurrences of Spasticity during Gait in Children with Cerebral Palsy

**DOI:** 10.3390/s24010041

**Published:** 2023-12-20

**Authors:** Axel Koussou, Raphaël Dumas, Eric Desailly

**Affiliations:** 1Pôle Recherche & Innovation, Fondation Ellen Poidatz, 77310 Saint-Fargeau-Ponthierry, France; axel.koussou@fondationpoidatz.com; 2Laboratoire de Biomécanique et Mécanique des Chocs UMR T9406, University Lyon, University Gustave Eiffel, University Claude Bernard Lyon 1, 69622 Lyon, France; raphael.dumas@univ-eiffel.fr

**Keywords:** spasticity, gait, muscle length, muscle velocity, muscle acceleration, electromyography

## Abstract

Spasticity might affect gait in children with cerebral palsy. Quantifying its occurrence during locomotion is challenging. One approach is to determine kinematic stretch reflex thresholds, usually on the velocity, during passive assessment and to search for their exceedance during gait. These thresholds are determined through EMG-Onset detection algorithms, which are variable in performance and sensitive to noisy data, and can therefore lack consistency. This study aimed to evaluate the feasibility of determining the velocity stretch reflex threshold from maximal musculotendon acceleration. Eighteen children with CP were recruited and underwent clinical gait analysis and a full instrumented assessment of their soleus, gastrocnemius lateralis, semitendinosus, and rectus femoris spasticity, with EMG, kinematics, and applied forces being measured simultaneously. Using a subject-scaled musculoskeletal model, the acceleration-based stretch reflex velocity thresholds were determined and compared to those based on EMG-Onset determination. Their consistencies according to physiological criteria, i.e., if the timing of the threshold was between the beginning of the stretch and the spastic catch, were evaluated. Finally, two parameters designed to evaluate the occurrence of spasticity during gait, i.e., the proportion of the gait trial time with a gait velocity above the velocity threshold and the number of times the threshold was exceeded, were compared. The proposed method produces velocity stretch reflex thresholds close to the EMG-based ones. For all muscles, no statistical difference was found between the two parameters designed to evaluate the occurrence of spasticity during gait. Contrarily to the EMG-based methods, the proposed method always provides physiologically consistent values, with median electromechanical delays of between 50 and 130 ms. For all subjects, the semitendinosus velocity during gait usually exceeded its stretch reflex threshold, while it was less frequent for the three other muscles. We conclude that a velocity stretch reflex threshold, based on musculotendon acceleration, is a reliable substitute for EMG-based ones.

## 1. Introduction

Cerebral palsy (CP) stands as the most prevalent motor disability in children, with 2–3 cases per 1000 live births worldwide [1]. This condition emanates from an upper motor neuron lesion that occurs in the early brain, leading to a set of neuromusculoskeletal impairments. Spasticity, often regarded as the most common primary impairment, is reported to occur in approximately 70–80% of children with CP [2,3]. Its assessment and treatment are, therefore, central to the management of a child with CP. The most accepted definition of spasticity refers to “a velocity-dependent increase in tonic stretch reflex with exaggerated tendon jerks, resulting from hyper-excitability” [4]. More recently, a European consensus stated that the term spasticity should only be used to define the “velocity-dependent neural part of hyper-resistance to stretch” [5]. Thus, spastic muscles are identified as hyper-reflexive to stretch: the reflex contraction in response to passive stretch is pathologically exaggerated due to a lack of central regulation, and depends on stretch velocity. This results in excessive and inappropriate muscle activation, which can contribute to functional impairments, including gait.

Clinically, an evaluator estimates the child’s spasticity by applying a quick stretch to a passive muscle and by assessing the resistance to the movement that results. Clinical scales, such as the Modified Tardieu Scale [6], have been widely used due to their ease of implementation. This scale involves two passive movements: one at a low joint speed (LV) and the other at a high joint speed (as fast as possible) (HV). The goal of the HV movement is to produce sufficient joint angular velocity in order to stretch the muscle–tendon unit complex and initiate the spastic muscle reflex. Then, the evaluator grades the severity of spasticity from no muscle reaction to the presence of a catch or clonus, or even a rigid joint. The joint angle at which the catch occurs is also measured. The difference between this catch angle during HV and the full passive range of motion during LV is known as the spasticity angle, with a greater range indicating a higher degree of spasticity. Although simple to implement, this technique has several limitations, including the inaccuracy of the catch angle measurement due to the subjectivity of the assessor. In this sense, the catch angle that is manually identified by the evaluator may not represent the true angle at initiation of muscle activation at a physiological level [2,7,8].

To improve the validity and reliability of spasticity evaluation, several authors have proposed instrumented assessments based on kinematics, kinetics, and electromyography (EMG) data [9,10,11,12]. Exaggerated muscle activity, measurable using surface EMG, during a passive muscle stretch can confirm the presence of spasticity. Then, combining kinematics, kinetics, and EMG recordings, several parameters (angle of catch, reflex moment, EMG bursts, etc.) can be determined to describe and quantify the subject’s spasticity [13]. Among these parameters, the velocity stretch reflex threshold, i.e., the velocity that triggers the stretch reflex, confirmed by an EMG burst, has been proposed and could be of great interest, particularly for comparing patients or for better analyzing the patient’s gait by detecting any occurrences of spasticity during this task [14].

Indeed, assessing the occurrence of spasticity during gait among children with CP remains challenging. While [4] provides a reasonably straightforward definition of spasticity when measuring a passive muscle, it might not apply to an active muscle that is voluntarily contracted, as in gait [15]. Therefore, different approaches have been proposed to detect spasticity and to evaluate its effect during gait [14,16,17,18]. One way is to detect EMG activities during gait shortly after some kinematic stretch reflex thresholds, generally velocity stretch reflex thresholds, previously established during passive assessment, have been exceeded [14]. This approach is convenient because it enables the straightforward assessment of the occurrence of spasticity during gait, i.e., when gait parameters are superior to the threshold values. However, these thresholds are determined through EMG-Onset (i.e., the beginning of the EMG burst) detection algorithms, which have been shown to have variable performance [19,20]. Indeed, EMG data are noisy and sensitive to electrode locations [21,22]. Moreover, the developed EMG-Onset detection algorithms depend on hyperparameters, which could be challenging to optimize for a large number of subjects. These reasons can lead to an unreliable determination of the EMG-Onset and, therefore, to an inconsistent threshold value. For example, if the EMG-Onset is determined too precociously or lately, the threshold values can be determined before the beginning of the stretch or after the kinematic or kinetic determination of the catch angle.

The passive stretch of a joint can be divided into several successive features of muscle–tendon kinematics: start, peak acceleration, peak velocity, peak deceleration, peak length, and stop. The possibility of using one of these features—that is, the patterns of the muscle–tendon length, velocity, and/or acceleration—to determine the stretch reflex threshold has not been evaluated yet. Recently, [23] showed that the timing of EMG-Onset was correlated with the stretch acceleration during stretches of the triceps surae, with higher accelerations seeming to evoke faster spastic responses (time to EMG-Onset). Similar findings have been made in previous studies showing a possible acceleration-driven stretch reflex activation in the triceps surae during postural responses. Specifically, it has been repeatedly found that acceleration is related to the EMG-Onset that occurs after muscle stretch due to standing perturbations [24,25,26].

As EMG-Onset might be difficult to determine with precision, especially in very fast movement such as high velocity stretches, it can be hypothesized that some features of the applied movement could be used as precise temporal markers to determine velocity stretch reflex thresholds, especially as some studies have shown correlations between applied acceleration and EMG response during passive stretches, including a correlation between EMG-Onset and stretch acceleration [23].

The aim of this study was, therefore, to evaluate stretch reflex thresholds determined from muscle–tendon acceleration timing in comparison to stretch reflex thresholds based on EMG-Onset in children with CP during passive muscle stretches. The stretch reflex threshold consistency was evaluated during passive stretches in terms of the timing of the reflex with respect to the beginning of the mobilization and with respect to the occurrence of the catch. The stretch reflex thresholds were also compared during gait in terms of the number of threshold exceedances, and of the proportion of the gait trial with a possible expression of spasticity.

## 2. Materials and Methods

### 2.1. Subjects

A total of 18 children with CP (mean age: 11.8 ± 2.8 years old; 12 males; Gross Motor Functional Classification Scale (GMFCS): I–III; 3 hemiplegic and 15 diplegic) were recruited from the Unité d’Analyse de Mouvement at Ellen Poidatz Fundation (Saint-Fargeau-Ponthierry, France) to participate in the study. Their characteristics are summarized in Table 1. The protocol was approved by the ethical committee of the Comité de Protection des Personnes—Ouest IV and informed consent was obtained from the children’s legal representatives (NCT04596852).

### 2.2. Experimental Protocol

The experimental procedure and its validity have already been presented in another study [27]. Briefly, the protocol consisted of two parts: a passive testing protocol and a gait analysis. Throughout the procedure, 3D body segment kinematics were determined at 100 Hz using a marker set placed over specific body landmarks (plug-in gait marker set with two additional markers placed over the iliac bone to take into account that the posterior superior iliac markers are not visible when the patient is lying on their back; for the passive testing protocol, the upper body markers were removed for the patient’s comfort) and a motion analysis system (15 cameras: 8 MX F20, 5 MX T40-S, 2 MX T160, VICON, Oxford, UK). EMG activities were synchronously recorded at 2000 Hz using pre-amplified dual differential surface electrodes (DE−2.1, DelSys, Inc., Boston, MA, USA) placed, bilaterally, over the rectus femoris (*Rf*), semitendinosus (*Sem*), soleus (*Sol*), and gastrocnemius lateralis (*Gas*) muscles. The electrode locations were determined according to the Surface Electromyography for the Non-Invasive Assessment of Muscles (SENIAM) guidelines and prepared by shaving the skin and cleaning it with alcohol.

Firstly, each child performed several gait trials at their self-selected speed along a 10 m walkway. Ground reaction forces were recorded at 2000 Hz using four embedded forceplates (2 AMTI, Watertown, MA, USA and 2 Kistler, Hampshire, UK).

Secondly, a passive testing protocol was performed to continuously measure the joint angle while the subject’s joint was manipulated as quickly as possible in order to elicit a stretch reflex response. The evaluator used a 3D handheld dynamometer (Sensix, Poitiers, France) to perform the subject’s joint mobilizations through the full available sagittal range of motion (ROM), while the participants were asked to relax. The passive joint moments were determined through inverse dynamics, using homogeneous matrices, from the joint kinematics and the forces and moments measured using the dynamometer [28,29]. The joints were mobilized three times in different supine positions to ensure the spastic characterization of the lower limb. To limit the experimental time, a choice was made to consider only four stretch positions and four muscles (Figure 1):Ankle with the knee at 90° and 0° (positions P1 and P2) for *Sol* and *Gas* evaluation;Knee with the hip at 90° and 0° (positions P3 and P4) for *Sem* and *Rf* evaluation.

Starting from a shortened position, the muscles of interest were lengthened at HV, toward ankle dorsiflexion, knee extension, or flexion for positions P1, P2, P3, and P4, respectively. Then, the return to the initial position was made at low velocity. Thus, only the lengthening phase was analyzed in this study. An interval of at least 5 s rest between repetitions was observed to avoid the effects of decreased post-activation depression in spastic muscles [30].

### 2.3. Data Analysis

*Sol*, *Gas*, *Sem*, and *Rf* muscle–tendon unit (MTU) length during mobilization or gait trials was computed via OpenSim by using the measured kinematics after model scaling [31]. The musculoskeletal model was also personalized by taking into account the child’s femoral and tibial torsions, measured by a clinician during the clinical evaluation, using the bone-deformation tool developed by [32]. All MTU lengths were expressed as a percentage of their MTU lengths in anatomical position. The MTU velocity (*v*_mt_, expressed in %/s) and acceleration (*a*_mt_, expressed in %/s^2^) were calculated using the derivative of the different MTU lengths. The signals were filtered after each derivation with a 6th-order zero-phase Butterworth lowpass filter at 15 Hz.

Raw sEMG signals were bandpass-filtered with zero-phase Butterworth filters (highpass 2nd order, 20 Hz and lowpass 8th order, 400 Hz). Then, EMG-Onset during high-velocity mobilizations was defined both visually (i.e., method Onset-Visu) or automatically (i.e., method Onset-Auto) [33]. This automatic algorithm, based on an approximated generalized likelihood principle, identifies onset as an abrupt change in the (time-varying) parameters of a statistical process model adapted to the measured signal and has been shown to perform significantly better compared to other algorithms [20,34]. Nevertheless, this algorithm can miss some EMG-Onsets when applied on noisy data. Thus, we reported the cases where no EMG-Onset was found using this automatic algorithm as missing values.

Then, the *v*_mt_ thresholds (*T*_vmt_) were defined in several ways (Figure 2): Onset-Visu: the *v*_mt_ at which EMG-Onset, determined visually, occurred minus 30 ms to consider the stretch reflex delay [35,36];Onset-Auto: the *v*_mt_ at which EMG-Onset, determined automatically [33], occurred minus 30 ms to consider the stretch reflex delay [35,36];MaxAcc: the *v*_mt_ at which the *a*_mt_ was maximal.

Finally, the *T*_vmt_ values, obtained with these three methods, were evaluated during passive stretch level or during gait. 

#### 2.3.1. Passive Stretch Analysis

First, we evaluated the consistency of the *T*_vmt_ during the passive stretch trial based on two criteria:The number of times the *T*_vmt_ was determined at a time before the mobilization had begun, i.e., pre-T0, where T0 defines the beginning of the stretch. These cases occurred when there was less than 30 ms between T0 and the EMG-Onset;The number of times the *T*_vmt_ was determined at a time after the occurrence of the catch, i.e., post-catch (determined as the maximal of the second derivative of the moment).

These two criteria represent physiological inconsistencies. The delays between T0 or the timing of the catch and the timings of EMG-Onset minus 30 ms or the maximum *a*_mt_ were also measured.

#### 2.3.2. Gait Analysis

Second, to determine the effect of a threshold’s difference, we computed the number of *T*_vmt_ exceedances followed by EMG activity in a window 20–90 ms after exceedance, i.e., where spasticity could be present. The phases of EMG activity during gait were determined with an automatic detection algorithm [37], which showed good performance in a pathological gait context, and were manually adjusted if necessary. Only the thresholds with no pre-T0 or post-catch inconsistency were analyzed. The window of 20–90 ms was chosen since the onset latency of the short-latency reflex is around 20–40 ms and the voluntary reaction time is around 100 ms [38,39,40]. The exceedance of the thresholds was not systematically followed by EMG activity. We reported these cases as false positive values. Then, concerning the threshold exceedances followed by EMG activity, we also reported the proportion of the gait trial time with *v*_mt_ > *T*_vmt_. Then, after checking the distribution of data (Shapiro–Wilk test), the values of *T*_vmt_, *T*_vmt_ exceedances, and proportion of the gait trial time with *v*_mt_ > *T*_vmt_ of the three methods were compared with nonparametric Wilcoxon signed-rank tests. The critical level of significance was set at *p* < 0.05 and adjusted with Bonferroni correction. Finally, we evaluated, for all methods, at which percentage of the gait cycle the *T*_vmt_ exceedances were present. These criteria represent clinical parameters of interest in a clinical context to evaluate whether spasticity is present during gait. The whole procedure is summarized in Figure 2.

## 3. Results

No EMG-Onset or clinical signs of spasticity were found for three, two, four, and three subjects in positions P1, P2, P3, and P4, respectively. This was due to the different muscle groups tested in the different positions, which may have had variable levels of spasticity across subjects. For each subject, the stretches were repeated three times, but some trials were excluded due to experimental issues (i.e., multiple occlusion of markers). In total 37, 42, 35, and 44 high-velocity stretches were analyzed for the muscles *Sol*, *Gas*, *Sem*, and *Rf*, respectively.

The mean maximal velocities (positive for joint flexion) were 287 ± 103°/s, 247 ± 66°/s, −331 ± 87°/s, and 231 ± 119°/s and the mean maximal acceleration (positive for joint flexion) was 4822 ± 2222°/s^2^, 4577 ± 1508°/s^2^, −2112 ± 829°/s^2^, and 1722 ± 973°/s^2^ for the positions P1, P2, P3, and P4, respectively. The mean maximal lengthening velocities were 71 ± 27%/s, 48 ± 12%/s, 56 ± 20%/s, and 33 ± 12%/s and the mean maximal lengthening acceleration was 1186 ± 527%/s^2^, 843 ± 262%/s^2^, 659 ± 331%/s^2^, and 328 ± 201%/s^2^ for the muscles *Sol*, *Gas*, *Sem*, and *Rf*, respectively.

### 3.1. Passive Stretch Analysis

Using the Onset-Auto method, we reported one, two, and one missing values for the muscles *Gas*, *Sem*, and *Rf*, respectively.

Table 2 presents the number of pre-T0 inconsistencies as well as the median (and interquartile range) delay between the beginning of the stretch (T0) and the EMG-Onset minus 30 ms or maximum *a*_mt_ value. Detailed results are available in the Appendix A.

Pre-T0 inconsistencies were found with both EMG-based methods, but not with the MaxAcc method. Except for *Sem*, median delays between T0 and the EMG-Onset minus 30 ms were found to be longer than between T0 and the maximum *a*_mt_ value.

Table 3 presents the number of post-catch as well as the median (and interquartile range) delay between the timing of the EMG-Onset minus 30 ms or maximum *a*_mt_ value and the catch. Detailed results are available in the Appendix A.

Post-catch inconsistencies were found with both EMG-based methods, but not with the MaxAcc method. Except for *Sem*, median delays between the EMG-Onset minus 30 ms and the catch were found to be shorter than between the maximum *a*_mt_ value and the catch.

### 3.2. Gait Analysis

Studying consistent *T*_vmt_ exceedances during gait showed that the false positive cases (*T*_vmt_ exceeded at least once during the gait trial without being followed by EMG activity) were frequent. *T*_vmt_ with at least one false positive case were found for the Onset-Visu, Onset-Auto, and MaxAcc methods, respectively, in position P1 (N = 9, 10 and 10 over 37 studied *T*_vmt_); P2 (N = 10, 14 and 14 over 42 studied *T*_vmt_); P3 (N = 19, 16 and 16 over 35 studied *T*_vmt_); and P4 (N = 5, 6 and 10 over 44 studied *T*_vmt_).

The descriptive parameters for the *T*_vmt_ values, Tvmt exceedances per gait cycle, and for the proportion of the gait trial time with *v*_mt_ > *T*_vmt_ values for the four tested muscles can be found in Table 4. Detailed results are available in the Appendix A. Five statistical differences between the methods were found: on *T*_vmt_ for *Sem* between the Onset-Visu and MaxAcc methods (*p* = 6.1 × 10^−4^) and between the Onset-Auto and MaxAcc methods (*p* = 0.001); on *T*_vmt_ for *Rf* between the Onset-Visu and Onset-Auto methods (*p* = 0.003) and between the Onset-Visu and MaxAcc methods (*p* = 0.006); and on the proportion of the gait trial time with *v*_mt_ > *T*_vmt_ for *Rf* between the Onset-Visu and Onset-Auto methods (*p* = 9.7 × 10^−4^). Comparative boxplots of the different parameters are available in the Appendix A.

For more than half of the *T*_vmt_ obtained during muscle stretches, the muscle lengthening velocity of *Sol*, *Gas*, and *Rf* during gait did not exceed it with following subsequent EMG activity, resulting in low *T*_vmt_ exceedances per gait cycle values. Moreover, when the threshold was exceeded and this was followed by EMG activity, it did not last long (low proportion of the gait trial time with *v*_mt_ > *T*_vmt_). Generally, the threshold exceedances occurred mainly at the beginning or at the end of the gait cycle (0–5% and 95–100%) for *Sol* (Figure 3—first row), at the beginning or during the swing phase of the gait cycle (0–10% and 70–100%) for Gas (Figure 3—second row), and at the beginning of the gait cycle (0–10%) for Rf (Figure 3—fourth row).

In contrast, the Sem muscle lengthening velocity during gait almost always exceeded, with subsequent EMG activity, its stretch reflex thresholds. For this muscle, the average value of *T_v_*_mt_ exceedance per gait cycle was one. The threshold exceedances occurred mainly during the swing phase (60–90% of gait cycle) (Figure 3—third row).

## 4. Discussion

The aim of this study was to evaluate whether the stretch reflex thresholds assessed in children with CP during passive muscle stretches, by the use of an instrumented assessment, could be determined from muscle–tendon kinematics instead of using EMG-based methods. The physiological consistency of the thresholds was assessed and we investigated the effect of the determined threshold on the estimated occurrence of spasticity during gait.

Missing and false positive values have also been reported, but are more considered as quality controls than as consistency tests. Indeed, thresholds are simply not computed when EMG-Onsets are not detected, and threshold exceedances during gait are not analyzed when they are not followed by any EMG activity. On the contrary, the number of times *T*_vmt_ was determined at a time before the mobilization had begun (pre-T0) and the number of times *T*_vmt_ was determined at a time after the occurrence of the catch (post-catch) have both been proposed in this study to evaluate the physiological consistency of the thresholds computed with the three methods. The method based on the maximum of the MTU acceleration was revealed to be the more consistent.

Statistical differences in the *T*_vmt_ values between the EMG-Onset and MaxAcc methods were only encountered for *Sem* and *Rf*. Nevertheless, once applied to gait, these *T*_vmt_ value differences did not lead to any statistical difference regarding the proportion of the gait trial time with *v*_mt_ > *T*_vmt_, or on the number of times where *T*_vmt_ was exceeded. This low number of statistical differences confirms that the MaxAcc method provides results close to the EMG-based ones. We would also like to highlight the fact that we found *T*_vmt_ values consistent with the study in [14] that tested *Gas* and *Sem* (median (IQR) values (%/s): *Gas*: 28.5 (9.8); *Sem*: 30.5 (15.5)). In addition, these authors also found that for about half of the children tested, the *Gas* muscle lengthening velocity during gait did not exceed its *T*_vmt_, while it was always exceeded for *Sem*. Concerning *Gas*, this finding could suggest that the abnormalities observed during gait could be more a consequence of non-neural hyper-resistance, i.e., higher passive stiffness, as suggested by other articles [41]. Thus, we believe that the proposed method based on maximal musculotendon acceleration could be a substitute for EMG-based ones. Even if, a priori, EMG-Onset methods are more physiologically relevant, they may lack reliability because the signals can be noisy and because it is difficult to precisely detect their abrupt change. Moreover, EMG-Onset detection algorithms depend on hyperparameters, which could be challenging to optimize for a large number of subjects. Indeed, the EMG-Onset methods have sometimes led to physiological inconsistencies. Taking the *v*_mt_ values 30 ms before EMG-Onset as *T*_vmt_, to take into account the stretch reflex delay admitted in the literature [35,36], led to some cases where the *T*_vmt_ values were determined at timings before the beginning of the stretch (pre-T0) or after the occurrence of the catch (post-catch). This was never encountered with the MaxAcc method. This method is based on a simple detection of a peak on kinematic curves. To obtain these curves, an instrumented assessment of the muscle stretches is required, with skin markers (or other sensors) allowing for musculoskeletal modeling. Nevertheless, as the objective is to compare *T*_vmt_ and *v*_mt_ during gait, gait analysis can be performed before muscle stretches, as in the present protocol, and the subjects are, therefore, already equipped with skin markers and sEMG electrodes.

We would also like to highlight the fact that we found electromechanical delays, i.e., the time between EMG-Onset and catch (median delay columns in Table 3; for EMG-Onset methods, the reader has to subtract 30 ms), to be more consistent with the literature [42,43] using the MaxAcc method than with the methods based on EMG-Onset detection. This is especially true for the muscles *Sol* and *Gas* where we found low mean electromechanical delays with Onset-Visu and Onset-Auto.

Some study limitations must be recognized. First, we chose some specific methods for EMG processing, automatic EMG-Onset, and catch determinations. Others methods have been proposed [11,19,37,44,45]. Using these methods, we might have found different results. Nevertheless, we found consistent electromechanical delay values that let us believe that the chosen methods are representative of the literature. Similarly, an interval of 5 s rest between stretch repetitions was observed to avoid the effects of decreased post-activation depression. However, it has been shown that post-activation depression can last up to 10 s [46]. For experimental reasons, we chose a slightly faster rest interval, as have other authors [30], and we consider that this has only a minor impact on the results obtained. Moreover, we used a non-specific value of 30 ms to represent the stretch reflex delay. However, this delay, which depends on body dimensions associated with the axon pathway to the target muscle, can vary among the subjects in a range of 23–33 ms for the soleus and 23–35 for the medial gastrocnemius [8]. Nonetheless, it remains experimentally difficult to assess subject-specific delay, as it requires the use of electrical nerve stimulation.

Second, we assumed that the evoked muscle EMG activity during passive HV stretches reflects involuntary reflex activation rather than voluntary muscle contraction. However, it is important to keep in mind that other feedback mechanisms, such as those via the supraspinal structures, and fast voluntary responses might also occur [23]. Despite this consideration, there are several reasons to assume that we are experiencing involuntary muscle responses. Indeed, explicit instructions were provided to the children to stay relaxed and we meticulously measured the background EMG before and throughout the applied stretch. Trials with higher levels of EMG activity before stretching were repeated. Furthermore, reinforcing the idea of the presence of abnormal activations, bursts in muscle responses were more frequent in children with CP compared to healthy ones (not included in this study), whereas voluntary anticipatory responses are expected to produce the opposite effect because it has been found, in patients with neurological lesions, that lowered sensory drive is required to produce these movements [23,47].

Third, for both the passive and gait conditions, although the population tested here involved children, we used an adult generic model that we scaled to the child’s height and modified according to the child’s femoral and tibial torsions. No more adaptations were made to the model to reflect pediatric or pathological muscle morphology. Moreover, we used this model to estimate muscle–tendon lengths even though, in the current paper, we refer to muscle lengths. Imaging data should be incorporated into future developments in order to upgrade the subject specification of musculoskeletal models. 

Fourth, an intra-subject variability of the kinematics and kinetics data between stretches could be present (Appendix A). This corresponds to the heterogeneity of the population (hemiplegic and diplegic children with GMFCS I to III). However, the triggering of the reflexes was consistent between the stretches, and even with this intrinsic variability, the MaxAcc method led to results similar to the EMG-based methods.

Fifth, the association between the spasticity assessed during passive muscle stretches and gait is complex. There is no consensus on whether spasticity assessed at rest reflects the activation of stretch reflexes during gait. Indeed, for the evaluation of the number of *T*_vmt_ exceedances, or the proportion of the gait trial time with *v*_mt_ > *T*_vmt_, we consider that any EMG activity following a threshold exceedance is a potential effect of spasticity. However, the measured activity could reflect non-spastic activation. For instance, *Gas* are usually activated during loading response, so the activity that we considered spastic might not be. We could have evaluated only the swing phase, as in [14], where muscles are supposed to be less active. But for children with CP, where EMG gait patterns are different, and where other muscle contractions, i.e., compensatory strategies or co-contraction, could be present, it would not have brought more clinical insights. Moreover, it has been demonstrated that stretch reflex thresholds can be modulated under voluntary movement. The authors in [48] showed that the stretch reflex occurred at larger joint angles during voluntary movement than during the passive mobilization of the elbows of post-stroke patients. The study in [49] also showed that the H-reflex was modified during gait in children with CP. In the current study, in the gait analysis section, we only reported the cases with subsequent EMG activity after *T*_vmt_ exceedance. However, we found several cases where *T*_vmt_ was exceeded without being followed by EMG activity (false positive values). This result would suggest that spasticity is not expressed during gait under the same velocity conditions as during the passive stretches in several patients, and could be explained by the fact that the mechanisms of spasticity during gait and during passive stretches are different [50]. In addition, we also found several cases where *T*_vmt_ was never exceeded during gait. Several hypotheses could explain these cases:Compensative gait adaptations in order to lower some muscles’ *v*_mt_ under their *T*_vmt_;Unsuitable motor schemes preventing walking at a *v*_mt_ above *T*_vmt_ (e.g., weakness of the agonists preventing the antagonists from being stretched too quickly that can be coupled with a higher antagonist’s passive stiffness, making the stretch more difficult; compensation due to a disorder at another joint; impaired selective motor control);*T*_vmt_ value above normal gait *v*_mt_, meaning that, in this case, spasticity has no impact on gait despite being triggered during instrumented physical examination.

Thus, this method could enable the ability to distinguish patients with a probable expression of spasticity during gait, under the same velocity conditions as during the passive stretches, and those with a probable absence of spasticity. However, while we are not able to determine whether a measured activity during gait is purely spastic, the full validity/accuracy (specificity and sensibility) of any method to determine stretch reflex thresholds cannot be evaluated. 

Finally, we only evaluated stretch reflex threshold on muscle lengthening velocity. Nevertheless, some studies suggest that spasticity could be due to other factors such as muscle length [51], muscle lengthening acceleration [23], or the force applied [52]. Thus, differences between both methods (EMG-Onset and MaxAcc) on these thresholds could be present and should be further evaluated. Thus, we might also have tested several velocities, not necessarily “*as high as possible*”, to further validate the method in comparison to the EMG ones. Still, using one velocity threshold as a marker of potential spasticity during gait remains a widely used method [14,16,53]. 

This study proposes a new method to determine *T*_vmt_ based on kinematics data. Larger studies are required to confirm these results, e.g., for other muscles or activities.

## 5. Conclusions

A kinematics method, based on maximal muscle–tendon acceleration, is proposed to determine MTU velocity stretch reflex thresholds. The proposed method produces results close to the EMG-based ones that are consistent with the literature. The proposed method is less sensitive to noisy data and provides no physiological inconsistency such as thresholds determined at a time before the mobilization or after the occurrence of the catch. For children with CP, a kinematics-based stretch reflex threshold could be a good substitute for EMG-based ones.

## Figures and Tables

**Figure 1 sensors-24-00041-f001:**
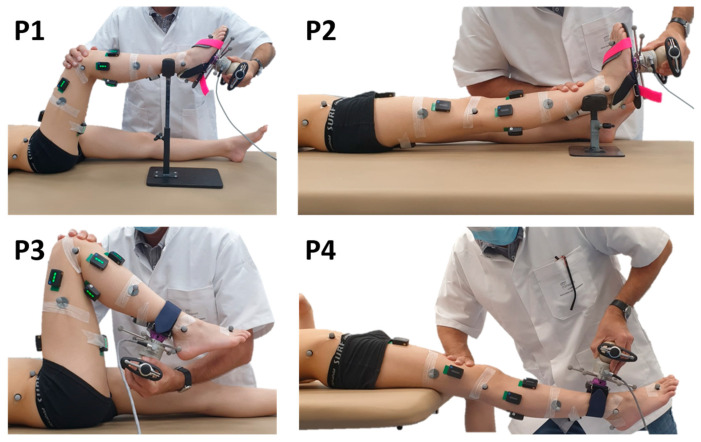
Test positions. The evaluator uses a 3D handheld dynamometer to mobilize joints through the sagittal range of motion. Dynamometer and body segment kinematics are measured with reflective markers and a motion analysis system. Muscle activities are measured with surface electromyography (sEMG).

**Figure 2 sensors-24-00041-f002:**
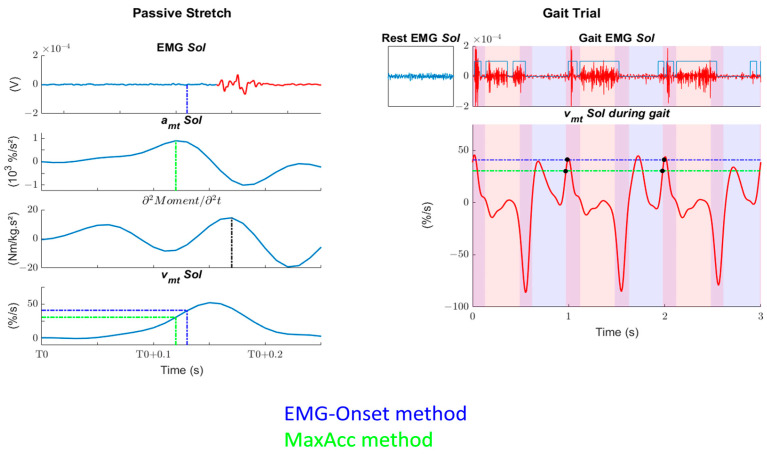
Example of data analysis of one subject after a passive stretch in position P1. Left: The figure represents the procedure of the musculotendon velocity threshold (*T*_vmt_) determination based on EMG (dot blue lines) or on kinematics (dot green lines). For ease of reading, only one EMG-Onset method is presented. First row: EMG recordings of the soleus muscle. Muscle activity is shown in red, while the blue dashed lines represent the EMG-Onset minus 30 ms. Second row: musculotendon acceleration of the soleus muscle. The green dashed lines represent the maximum. Third row: second derivative of the passive moment. The black dashed lines represent the maximum, which we consider to be the catch. Fourth row: musculotendon velocity (*v*_mt_) of the soleus (*Sol*) muscle. The green and blue dashed or dash-dotted lines represent the *T*_vmt_ determined through MaxAcc and EMG-Onset methods, respectively. T0: beginning of the stretch. Right: The figure represents the relation between *T*_vmt_ and gait data. First row: EMG recordings of the soleus (*Sol*) muscle at rest or during gait. Phases of activity during gait are determined automatically and checked visually. Second row: *v*_mt_ during gait is compared to *T*_vmt_ determined during passive stretches. The number of crossings followed by EMG activity (black dot) are counted, as well as the time that the *v*_mt_ stays above *T*_vmt_. Red shaded areas represent single-stance phases, blue areas are for swing phases, and their superposition represents double-support phases.

**Figure 3 sensors-24-00041-f003:**
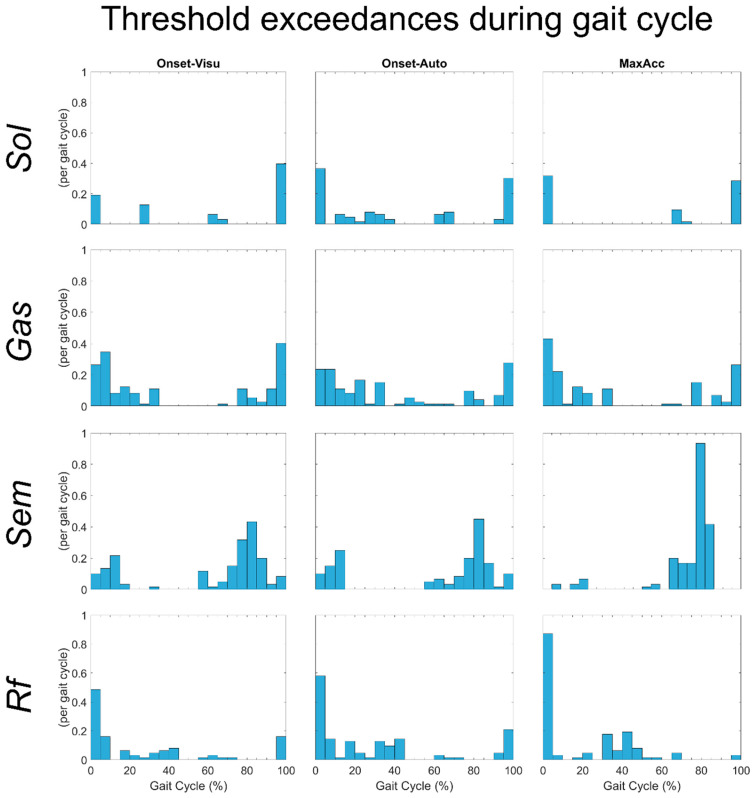
Histogram of the number of threshold exceedances during all gait trials of all subjects for the different muscles tested.

**Table 1 sensors-24-00041-t001:** Subject characteristics presented in correspondence with each muscle studied. *Sol*: soleus; *Gas*: gastrocnemius lateralis; *Sem*: semitendinosus; *Rf*: rectus femoris. Due to some experimental and processing/modeling issues, the number of trials (*n*) is different for each muscle and, therefore, the corresponding subjects’ characteristics. GMFCS: Gross Motor Functional Classification Scale. MTS: Modified Tardieu Scale—dorsiflexion ankle angle for *Sol* and *Gas* muscles; knee flexion angle for *Sem* and *Rf* muscles.

Subjects	*Sol* (*n* = 37)	*Gas* (*n* = 42)	*Sem* (*n* = 35)	*Rf* (*n* = 44)
Gender m/f	11/4	10/6	9/5	9/6
Mean age (SD)	13.4 (2.4)	13.3 (2.4)	13.3 (2.6)	12.9 (2.3)
GMFCS	I: 5 II: 8 III: 2	I: 4 II: 9 III: 3	I: 3 II: 9 III: 2	I: 4 II: 8 III: 3
Mean MTS angle (SD)	−6.0 (13.5)	−15.6 (8.7)	78.2 (11.3)	72.6 (40.5)

**Table 2 sensors-24-00041-t002:** Number of pre-T0 inconsistencies and delay (in ms) between the beginning of the stretch (T0) and the EMG-Onset minus 30 ms or maximum *a*_mt_ values for the different muscles.

	Onset-Visu		Onset-Auto		MaxAcc	
	pre-T0	Median delay [IQR]	pre-T0	Median delay [IQR]	pre-T0	Median delay [IQR]
*Sol*	0	93.5 [51.5–117.0]	3	76.5 [48.0–108.0]	0	60.0 [40.0–90.0]
*Gas*	2	75.2 [55.9–104.6]	4	77.0 [52.0–99.5]	0	60.0 [40.0–80.0]
*Sem*	0	183.5 [132.5–205.3]	3	175.0 [118.5–238.0]	0	190.0 [140.0–250.0]
*Rf*	0	222.5 [173.3–274.0]	1	197.0 [150.8–271.3]	0	180.0 [140.0–232.5]

**Table 3 sensors-24-00041-t003:** Number of post-catch inconsistencies and delay (in ms) between the timing of the EMG-Onset minus 30 ms or maximum *a*_mt_ values and the catch for the different muscles.

	Onset-Visu		Onset-Auto		MaxAcc	
	pre-T0	Median delay [IQR]	pre-T0	Median delay [IQR]	pre-T0	Median delay [IQR]
*Sol*	0	93.5 [51.5–117.0]	3	76.5 [48.0–108.0]	0	60.0 [40.0–90.0]
*Gas*	2	75.2 [55.9–104.6]	4	77.0 [52.0–99.5]	0	60.0 [40.0–80.0]
*Sem*	0	183.5 [132.5–205.3]	3	175.0 [118.5–238.0]	0	190.0 [140.0–250.0]
*Rf*	0	222.5 [173.3–274.0]	1	197.0 [150.8–271.3]	0	180.0 [140.0–232.5]

**Table 4 sensors-24-00041-t004:** Median [and inter-quartile range] of the *v*_mt_ thresholds (*T*_vmt_, expressed in %/s), the number of times, by gait cycle, that *T*_vmt_ was exceeded and followed by EMG activity and the proportion of the gait trial time with visually confirmed EMG activity where *v*_mt_ > *T*_vmt_, i.e., where spasticity could be present for the three methods.

	*T* _vmt_			*T*_vmt_ Exceedances per Gait Cycle			% of Gait Trial Time with *v*_mt_ > *T*_vmt_		
	Onset-Visu	Onset-Auto	MaxAcc	Onset-Visu	Onset-Auto	MaxAcc	Onset-Visu	Onset-Auto	MaxAcc
*Sol*	37.6 [25.9–66.8]	32.9 [24.2–62.3]	38.3 [26.6–53.8]	0 [0–0.25]	0 [0–1.00]	0 [0–0.80]	0 [0–0.5]	0 [0–2.1]	0 [0–2.6]
*Gas*	20.8 [14.1–31.0]	31.8 [24.6–37.4]	27.4 [22.5–32.1]	0.67 [0–1.00]	0 [0–1.00]	0.25 [0–1.00]	1.2 [0–8.1]	0 [0–5.7]	0.4 [0–3.5]
*Sem*	17.5 [10.8–24.4]	17.2 [14.2–19.7]	38.3 [27.9–42.8]	1.00 [0.33–1.00]	0.78 [0.38–1.00]	1.00 [0–1.00]	7.8 [3.0–12.0]	6.6 [4.1–10.5]	5.8 [0–9.1]
*Rf*	25.6 [19.6–31.2]	22.1 [16.5–31.7]	20.2 [16.4–25.2]	0 [0–0.75]	0.38 [0–1.00]	0.25 [0–1.00]	0 [0–1.8]	0.6 [0–5.1]	0.2 [0–6.5]

## Data Availability

The data presented in this study are available on request from the corresponding author. The data are not publicly available due to privacy and ethical restrictions.

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
