# Peer review of "A Velocity Stretch Reflex Threshold Based on Muscle–Tendon Unit Peak Acceleration to Detect Possible Occurrences of Spasticity during Gait in Children with Cerebral Palsy"

_sensors, 2023, doi:10.3390/s24010041_

Round 1

Reviewer 1 Report

Comments and Suggestions for Authors

The authors present experimental work and analysis targeted at systematically identifying spasticity-related muscle responses to rapid passive stretches of leg muscles as well as in gait. The work is targeted at improving the quantification capabilities for spasticity in children with cerebral palsy. The novel proposed method relies on the second derivative (acceleration) of the muscle-tendon unit length calculated from an OpenSim model and is more reliable than previous measures based on EMG.

I must state that I am not an expert in the field as I have never done a single spasticity evaluation with a patient myself. However, I am familiar with the disease and its implications on gait. So I am very interested in the subject and outcomes. But I must state that in its current form it took me a lot of effort to understand what is happening here. The core evaluation details seem implicit in the methods and I needed to go back and forth to intro and figures to put it together, hopefully.
This may all very well be easily accessible to an expert, but unfortunately not to me. I will detail below my difficulties and would suggest a revision to make the methods more accessible to a broader readership.

Did I get the core idea right?
- The movement is applied externally by the therapist together with the dynamometer
- The external movement sometimes causes spasticity, occurring as a "catch" before the end of the slow passive range of motion
- You measure the catch by taking the second derivative of the moment

What now? What is the relation between measured threshold times and the catch? Why does the peak amt contain any information on the spasticity? Is this not simply a measure of the movement induced by the therapist?

What are the independent values here? Do you vary the velocity of the movement to see at which velocity the spasticity is triggered? Or is it simply determined in the single movement and the velocity at Tvmt is then the threshold which is used in the gait analysis? This is what I think I understood from the figure.

I understand how Tvmt are determined, both methods.

Please try to write this in a consistent description in the methods section.

Other:

The introduction is mostly well written and gives a good overview of the field and problem. However, it is not sufficiently clear to me from the introduction what the concept "threshold based on muscle-tendon acceleration" actually means. The following was my problem when I first red it: Movement is applied externally to the leg. EMG measures the muscle activity and if it exceeds a certain level, this indicates the onset of the spasticity and therefore the threshold. But how does this translate to acceleration, which is part of the cause of the movement, not an effect of the movement. Maybe add a bit more to make this clear already in the last paragraph of the introduction to allow the reader to easily grasp the concept. Maybe even add a simple explanatory figure?

- Line 148/149: How did you deal with the reaction forces of the table in the inverse dynamics?
- 175: cloud you please motivate the 15 Hz?
- To me, vmt and Tvmt and similar are variables and should be written with the main characteristic as an italic letter and respective indices. The current notation is confusing as it matches e.g. also the way you name the methods etc.
- L 188: please make three bullet points out if these and put the names of the definitions first. E.g.: - MaxAcc: v_mt at which a_mt was maximal (or similar)
- L 194: with the two levels, you mean two speeds or passive stretch and gait analysis?
- L 259 (and elsewhere: please consider revising the presentation of the numbers with less significant digits. It makes in my opinion no sense to report five significant digits (see eg.g. https://web.ics.purdue.edu/~lewicki/physics218/significant or https://en.wikipedia.org/wiki/Significant_figures#Writing_uncertainty_and_implied_uncertainty)
- L239: could you please elaborate a bit why you took the second derivative of the passive moment to determine the catch? What is the reasoning.

Questions our of curiosity:
- there is some common background here with other spastic diseases like hereditary spastic paraplegia, where also the upper motor neuron is affectd. Consequences are spasticity and muscle weakness. Do you expect that your method could also work there`
- Do you think that the threshold changes in neurodegenerative diseases with disease progression?

Author Response

Dear Reviewer, 

Thank you for your expertise. 

A point-by-point response to your comments should be available in the attachments. 

Thanks you in advance,

Regards

Reviewer 2 Report

Comments and Suggestions for Authors

Thank you for an opportunity to review the manuscript entitled "A velocity stretch-reflex threshold based on muscle-tendon unit peak acceleration to detect possible occurrences of spasticity during gait in children with cerebral palsy".

The manuscript aimed to evaluate the feasibility of determining velocity stretch-reflex threshold, from the maximal musculo-tendon acceleration instead of the based on EMG method.

Generally, the justification of the study is clearly stated and well written. Methods is concise and easy to follow. However the characteristics of 18 children was mentioned that they were classified as GMFCS levels I to III, which could result in the variability between participants.

Discussion could be improved by mentioning about the heterogenous of the participants. children with spastic hemiplegic cerebral palsy are different from children with spastic diplegic cerebral palsy, especially when they are walking. How this study overcome this limitation.

Author Response

(The authors gave the same response as above.)

Reviewer 3 Report

Comments and Suggestions for Authors

Please find some remarks in the attached file.

Author Response

(The authors gave the same response as above.)

Round 2

Reviewer 1 Report

Comments and Suggestions for Authors

Thank you for your detaild responses. The approach is now clearer and from the other reviewers I learned that this is also clear to them. Therefore I can accept the manuscript in its current form.